# Factors Associated with Potentially Inappropriate Screening for Vitamin D Deficiency among Women in Medically Underserved Regions of West Texas

**DOI:** 10.3390/jcm12030993

**Published:** 2023-01-28

**Authors:** Duke Appiah, Samira Kamrudin, Cornelia de Riese

**Affiliations:** 1Department of Public Health, Texas Tech University Health Sciences Center, Lubbock, TX 79430, USA; 2Department of Obstetrics & Gynecology, Texas Tech University Health Sciences Center, Lubbock, TX 79430, USA

**Keywords:** vitamin D, women, medical overuse, vitamin D deficiency, medically underserved area, Texas

## Abstract

Testing for vitamin D deficiency (VDD) has been on the increase due to its association with several diseases. However, inappropriate testing for VDD, defined as screening for VDD among individuals with a low risk, has been reported. The aim of this study was to evaluate the prevalence and factors associated with potentially inappropriate screening for VDD among medically underserved populations in West Texas. Data were from 21,407 women who were hospitalized from 2016 to 2018 at a large regional health system. Logistic regression was used to calculate odds ratios (ORs) and 95% confidence intervals (CIs). The median age of patients was 40 years. While the proportion of patients tested for VDD reduced from 8.9% to 7.6% (*p* = 0.013) from 2016 to 2018, the prevalence of inappropriate testing increased from 32.3% to 46.8% (*p* < 0.001), with the 3-year prevalence of VDD being 27.6%. White race (OR = 2.71, CI: 1.95–3.78), an age ≥ 65 years (OR = 3.07, CI: 2.05–4.59), the use of public-sponsored insurance (OR = 1.62, CI: 1.20–2.17), cardiovascular disease (OR = 0.75, CI: 0.63–0.90), and vitamin D supplement use (OR = 7.05, CI: 5.82–8.54) were associated with inappropriate testing for VDD. In this study, an increasing prevalence of potentially inappropriate testing for VDD was observed. Sociodemographic and health-related conditions were associated with potentially inappropriate testing for VDD.

## 1. Introduction

Vitamin D is known to play essential roles in bone and mineral metabolism [1]. Several reports, mainly from observational studies, have suggested that vitamin D deficiency (VDD) contributes to the development of numerous chronic conditions, including osteoporosis [2], colorectal cancer [3], breast cancer [4], cardiovascular disease [5] and metabolic disorders [6], as well as overall mortality [6]. However, evidence from clinical trials shows little to no association between vitamin D supplementation and several of the aforementioned health conditions [7,8,9,10,11,12]. Gender differences in vitamin D status have been reported [13,14]. Although not conclusive, some studies have reported higher rates of VDD among women, especially post-menopausal women [2,15]. Furthermore, a high prevalence of VDD has been observed among uninsured and medically underserved female populations [16].

The awareness of vitamin D’s possible association with several diseases has resulted in increased testing for VDD among all age groups in the United States and several parts of the world over the past two decades [1,17,18,19]. Women are often more likely to be tested than men [19,20]. There is currently no evidence to demonstrate benefits for screening for VDD using serum 25-hydroxyvitamin D, a biochemical measure of vitamin D status, in the general population [21,22,23]. To limit wasting resources and unnecessary treatment in healthy individuals, testing for VDD is only recommended for individuals at a high risk for VDD [24]. These include persons with any of the following conditions: bone metabolic pathology; chronic kidney disease; hepatic failure; malabsorptive syndromes such as cystic fibrosis, coeliac disease, Crohn’s disease, gastric bypass surgery, and radiation enteritis; hyperparathyroidism; granuloma-forming disorders; or persons taking medications such as glucocorticoids, anticonvulsants, or highly active antiretroviral therapy [1,21,24,25]. Vitamin D screening is also indicated for pregnant and lactating women, individuals with obesity, persons with dark skin, people with Hispanic ethnicity, and older adults with a history of falls and nontraumatic fractures [1,24].

Despite recommendations against screening for VDD in low-risk individuals, high rates of vitamin D testing persist in this population [24]. About 25% to 78% of all vitamin D tests are deemed to be inappropriate (i.e., nonindicated or unnecessary), which is defined as screening for VDD among individuals with a low risk [26,27,28,29,30]. Inappropriate testing for VDD often leads to unnecessary treatment in a significant subgroup of healthy individuals [24,26]. Although more than two-thirds of women from medically underserved areas are reported to have VDD [16], there are limited investigations on inappropriate testing for VDD deficiency in this population.

The medical care of vulnerable and underserved populations is of great importance to public health. Therefore, the aim of this study was to evaluate the prevalence and salient factors associated with potentially inappropriate screening for VDD among medically underserved populations in West Texas, a region with a population of about 2.3 million people consisting of 35% White, 57% Hispanic, and 4.3% Black or African American persons [31]. All 70 counties in West Texas with shortage designation data are classified as either medically underserved areas or medically underserved populations, which are defined as geographic areas or populations with a lack of access to primary care services and facing economic, cultural, or linguistic barriers to access health care [32].

## 2. Materials and Methods

### 2.1. Study Population

Data for this study were obtained from women who were admitted from 2016 to 2018 into a large regional health system in Lubbock County. This county-owned, non-profit health system provides comprehensive healthcare to individuals living in West Texas and Eastern New Mexico. The current study was based on data from 21,407 women who were admitted from 2016 to 2018 and consented to their information being used for research purposes. Data collection protocols were approved by the Institutional Review Board of the Texas Tech University Health Sciences Center (IRB #: L19-100).

### 2.2. Definition of Variables

The following variables were extracted from electronic health records: age, sex, race, insurance status, marital status, body mass index, date of admission, date of discharge, length of stay, in-hospital mortality, discharge diagnosis for the principal condition, and two other secondary conditions: vitamin D supplement intake and vitamin D tests at first admission and the results of the test. No data on current prescription medications taken by patients were extracted from the health records.

VDD was defined as 25-hydroxy vitamin D levels of <12 ng/mL (30 nmol/L) [33,34,35]. Obesity was defined as a body mass index ≥ 30 Kg/m^2^. Prevalent medical conditions were defined using the tenth version of the International Classification of Disease (ICD-10). Discharge codes I00–I99 were used to define cardiovascular disease (CVD), C00–C97 for cancer, E11 for type 2 diabetes, and J40–J44 for chronic obstructive pulmonary disease. Patients were considered to be at a high risk for VDD if they reported Black race or Hispanic ethnicity, were obese, or reported any of the following conditions: pregnant and/or lactating (Z33, Z34, O00–O9A); rickets (E55); osteomalacia (M83); osteoporosis (M81); adults aged > 50 years with history of falls or nontraumatic fractures (W18.30XA, Z91.81); chronic kidney disease (N18, N19); liver disease (K70-K77); pancreatic insufficiency (K86); malabsorptive syndromes (V44, K90) such as cystic fibrosis (E84); inflammatory bowel disease (K50, K51); Crohn’s disease (K50); celiac disease (K90.0); gastric bypass surgery (Z98.84) and radiation enteritis (K52.0, K62.7); hyperparathyroidism (E21.2); lymphoma (C81–C88); and granuloma-forming disorders, namely sarcoidosis (D86), tuberculosis (A15–A19), histoplasmosis (B39), coccidiomycosis (B38), and berylliosis (J63.2) [21,24,25]. Vitamin D testing among women who were not considered to be at a high risk for VDD were classified as a potentially inappropriate screening for VDD.

### 2.3. Statistical Analysis

Characteristics of patients at the first visit as well during the hospital stay were described according to vitamin D testing status and whether the tests performed were potentially inappropriate or not using the Chi-square test and Fisher’s exact test. To evaluate the factors associated with inappropriate testing among patients with no high-risk conditions for VDD, multivariable logistic regression models were employed to calculate the odds ratios (ORs) and 95% confidence intervals (CIs). The sociodemographic and health-related factors considered for the multivariable model were age (18–34, 35–64, or ≥65 years), race (White or non-White), marital status (single, married, divorced/separated/widowed or other/unknown), type of insurance (private, public, or uninsured), vitamin D supplement intake (yes or no), CVD (yes or no), cancer (yes or no), and diabetes (yes or no). These variables were selected because they have been reported to be associated with vitamin D status in the literature [2,3,4,5,6] and were among the variables available for analysis in the current study. Also, in bivariate analyses, all these factors were significantly associated (*p* < 0.05) with the odds of testing for VDD. In all analyses, statistical significance was defined based on a two-tailed alpha value of 0.05, with SAS software version 9.4 (SAS Institute, Inc., Cary, NC, USA) used for data analyses.

## 3. Results

The median age of the 21,407 women was 40 (interquartile range: 28–64) years with 42% being less than 35 years old. Approximately 83% of participants reported White race with 6.4% reporting Black race. More than half (60%) of the women used public-sponsored insurance programs, with the majority of patients admitted in 2016. At the time of admission, 7.1% of the patients were taking vitamin D supplements. With regard to chronic health conditions, 26.9% had a discharge diagnosis for cardiovascular disease, while 7.4% and 4.2% had discharge diagnoses for diabetes and cancer, respectively. More than a third of the women (38.2%) were either pregnant or lactating at the time of admission. Almost 74% of the women admitted had a medical condition or race and ethnicity that were considered to be an indication for a high risk for VDD.

From 2016 to 2018, the proportion of patients tested for VDD reduced from 8.9% to 7.6% (*p* = 0.013). The prevalence of VDD over the study period reduced from 33.8% to 18.5% with the 3-year prevalence being 27.6% (Figure 1). The prevalence of VDD was lowest among White women (26.4%) compared with Black (36.6%), Hispanic (32.1%), and women who reported “Other” race (33.0%) (*p* = 0.050). Characteristics of participants according to vitamin D testing are reported in Table 1. Women who were older; reported White race; used public-sponsored insurance programs; were either divorced, separated or widowed; were taking vitamin D supplements; or had chronic diseases, namely CVD, diabetes, cancer, and chronic obstructive pulmonary disease tended to be tested more often for VDD than women without these conditions.

The overall prevalence of potentially inappropriate testing was 39.6%, increasing from 32.3% in 2016 to 46.8% in 2018 (Figure 1). The characteristics of participants according to potentially inappropriate testing for VDD are shown in Table 2. The prevalence of VDD was higher among high-risk women who were tested for VDD compared with women who received inappropriate testing (33.4% vs. 18.8%). Similarly, in-hospital mortality was greater among high-risk women who were tested for VDD compared with women who received potentially inappropriate testing (15.6% vs. 7.9%).

In multivariable adjusted models (Table 3), prior Vitamin D supplement intake (OR = 7.05, 5.82–8.54), older age (OR = 3.07, 2.05–4.59), White race (OR = 2.71, 1.95–3.78), and the use of public-sponsored insurance programs (OR = 1.62, 1.20–2.17) were positively associated with potentially inappropriate testing for VDD, while women with CVD had 25% lower odds of potentially inappropriate testing for VDD (OR = 0.75, 0.63–0.90). Marital status, diabetes, and cancer were all not significantly associated with potentially inappropriate testing for VDD.

## 4. Discussion

In this study of a large population of women residing in medically underserved regions of West Texas from 2016 to 2018, the proportion of patients tested for VDD reduced over time. However, the prevalence of potentially inappropriate testing for VDD deficiency increased over the same time period. Prior vitamin D supplement intake, older age, White race, the use of public-sponsored insurance programs, and prevalent CVD were all significantly associated with potentially inappropriate testing for VDD. To our knowledge, this study, which was the first to evaluate the inappropriate testing of VDD in medically underserved populations, showed that several subgroups within this population were at risk for more testing for VDD, despite being at a low risk for VDD.

Although routine vitamin D testing is not recommended for low-risk populations by several medical organizations, including the U.S. Endocrine Society, the National Academy of Medicine, and the U.S. Preventive Services Task Force, as well as advocacy groups such as the Choosing Wisely Initiative [25,36,37], there are still widespread reports of increased testing for VDD in several parts of the world [1,17,18,19,38]. This is primarily due to reports of associations of vitamin D levels with several infectious and chronic diseases as well as excess mortality from observational studies [1], although evidence from experimental studies have reported little to no benefit of vitamin D supplementation on these outcomes [7,8,9,10,11,12]. VDD has mainly been driven by insufficient exposure to sunlight and insufficient consumption of vitamin-D-rich foods [39]. Despite the definition and relevance of VDD being under debate, numerous studies have reported a substantial prevalence of VDD in several parts of the world [39]. In the United States, the prevalence of VDD has been reported to range from 2.6% to 29% [34,40,41], although estimates among uninsured and medically underserved female populations have been higher [16]. Accordingly, the prevalence of VDD in the current study was substantially higher (27.6%) than those reported among the general female population in the United States by Herrick et al. [34], who used the same cutoff value of 25-hydroxyvitamin D levels of <12 ng/mL (30 nmol/L) that was used to define VDD in the current study. These results were also reflective of the fact that VDD among hospitalized patients often tends to be higher than outpatients or the general population [29]. The estimate of VDD in the current study was similar to the reported prevalence of VDD of 21% among underserved women attending a county-sponsored free medical clinic in urban Michigan [16]. As expected, the prevalence of VDD in the current study was highest among Black women, an observation that corroborated several reports that show that Black men and women are more vulnerable to VDD than other races due to dark skin pigmentation inhibiting the skin synthesis of vitamin D [16].

The widespread report of VDD in observational studies has also resulted in increasing rates of the inappropriate or unnecessary testing of vitamin D levels in the clinical setting [20,27,28]. Despite women from medically underserved populations having a high prevalence of VDD, there is limited evidence of inappropriate testing in this population. The current study showed that the prevalence of inappropriate testing of VDD among medically underserved women from West Texas increased from 32.3% in 2016 to 46.8% in 2018. In other patient populations who are not from medically underserved areas, the reported prevalence of unnecessary testing for VDD has mostly been higher than those reported in the current study. In Spain, the retrospective analysis of vitamin D tests conducted at a high-level complexity center from 2009 to 2014 showed that 25% of all tests were clinically or biochemically unjustified [29]. Using a representative sample of requested vitamin D tests conducted at a tertiary healthcare center in Croatia during the year of 2018, Aralica et al. [27] reported that 57% of all tests were unnecessary according to national guidelines. The highest proportion of vitamin D tests considered inappropriate reported to date came from an evaluation of data on vitamin D testing at a healthcare center in the United Kingdom during 2017 by Woodford et al. [18], who reported that as much as 77.5% of vitamin D testing was potentially inappropriate.

Only a few studies have evaluated factors associated with inappropriate testing for VDD. In the current study, several demographic, socioeconomic, and health factors, namely older age, White race, the use of public-sponsored insurance programs, having a prevalent chronic disease like cardiovascular disease, and vitamin D supplement intake were all significantly associated with potentially inappropriate testing for VDD. On the one hand, some of these results supported findings from some prior studies on inappropriate testing for VDD [20]. For instance, a greater proportion of patients at a primary care center of a large regional health system in Southwest Virginia, who had inappropriate testing for VDD, were White persons [26]. Older age has also been reported to be associated with more inappropriate testing for VDD [20,28]. On the other hand, some of these findings for associated factors found in the current study were in contrast with results from other studies [26]. Specifically, Rockwell et al. reported that patients who received inappropriate testing for VDD were younger, more likely to be commercially insured or self-paid, and less likely to be insured by Medicare compared with patients with adequate vitamin D testing [26]. Besides the differences in socioeconomic status and types of patients (primary care vs. hospitalized) between the current study and that of Rockwell et al., the latter study excluded patients with inappropriate testing who were already taking vitamin D supplements [26], a factor that was strongly associated with inappropriate testing in the current study. With more than a third (37%) of Americans over 60 years of age reported to take vitamin D supplements, the highest of any age group [42], the exclusion of such patients by Rockwell et al. resulted in a relatively younger population who were also less likely to be on public-sponsored insurance programs [26].

While vitamin D tests are relatively cheaper than other biochemical tests, costing about $100 to $300 per test, the large volume of vitamin D tests often conducted on a regular basis makes it a significant economic investment [26,43]. A recent report noted that 35% of tests for VDD conducted in the state of Washington were unnecessary, resulting in an estimated cost of $12 million [44], while unnecessary testing for VDD was reported to cost $9.6 million in the state of Maine and over $20 million in the state of Virginia (thus 0.9% of the state’s healthcare spending in 2014) [43,45]. Inappropriate testing for VDD is often more common among Medicare patients than commercially insured patients [43,46]. In the current study, almost 60% of the entire population of women, as well as 76% of those who had potentially inappropriate vitamin D tests, reported using public-sponsored insurance programs. These public programs often cater to low-income populations. Taken together, the finding that as much as 47% of all vitamin D tests in the current study are potentially inappropriate is of significant importance to medically underserved populations, since it places a substantial financial burden on the already strained healthcare system in West Texas. Besides the financial burden, unnecessary testing for VDD is considered a source of low-value health care [26,37], with some studies reporting a downstream health service cascade involving unneeded additional vitamin-D-relevant laboratory testing, the ordering of prescriptions, and imaging services [26]. All these can lead to patient discomfort, patient harm, and an unnecessary additional financial burden for both patients and the healthcare system [47].

Emerging results from studies using different strategies to limit unnecessary testing for VDD have largely shown positive results [20,29,48,49,50]. An intervention in Australia using more restrictive criteria for testing, whereby benefits were only paid by the Medicare Benefits Schedule when vitamin D testing was performed among high-risk groups, resulted in a 47% decrease in vitamin D testing [20]. However, the proportion of tests with no indication increased from 71.3% to 76.5%, with practices located in high socioeconomic areas continuing to report the highest rates of testing [30]. In the U.S, the implementation of a decision-support tool in electronic medical records of a large health system resulted in a decrease in inappropriate testing from 43.8% to 30.3% (6 months after intervention) [48]. In Alberta, Canada, Naugler et al. [49] reported that a provincially led intervention based on the Choosing Wisely Canada recommendation resulted in a large and sustained reduction (91.4%) in serum total 25-hydroxyvitamin D testing over the period of one year. However, only 45% of primary care physicians in this province supported the specialized test requisitions [51]. For the most part, the findings of Naugler et al. provided evidence that county- or district-wide interventions involving the broad engagement of key stakeholders such as clinical laboratories, public health departments, and medical associations may offer a better avenue to reduce inappropriate testing for VDD, especially if they do not offer patients the option to pay for non-indicated testing [49]. Until that is achieved in medically underserved areas, the continued education of physicians about the appropriate use of laboratory tests and the frequency of testing, in addition to educating patients using recommendations from Choosing Wisely, will all go a long way to reduce unnecessary testing for VDD [20].

The strength of the current study included the use of a large population of women from medically underserved areas. Furthermore, the short period of observation (3 years) of the current study limited the impact of changes in unmeasured factors, such as the availability and marketing of tests, as well as changes in population preferences for some tests [28]. The findings of the current study, however, should be interpreted in light of the following limitations. The study was based on electronic health records, and research with such records makes assumptions about clinician behavior without necessarily knowing intent [26]. Thus, they also do not capture details that may have factored into a clinician’s decision to conduct vitamin D testing [26]. Because information on prescription and over-the-counter medication use were not available, the inappropriate testing for VDD was mainly based on health conditions that may not have all been accurately captured by ICD codes. This may have resulted in some tests that were classified as inappropriate being actually indicated and vice versa. However, some professional organizations do not include information on prescription medication in the determination of inappropriate testing [37]. Data on the specialty of the physicians requesting the tests, as well as the analytic methods used to quantify vitamin D levels, were not available. The capturing of vitamin D supplement intake may not have been complete since over-the-counter vitamin D supplement use was only known if reported by patients. Finally, the data were from hospitalized women and may or may not be generalizable to all women from medically underserved regions of West Texas.

## 5. Conclusions

In conclusion, the results from this study of women living in medically underserved regions of West Texas showed an increasing prevalence of potentially inappropriate testing for VDD over time. Sociodemographic and health-related conditions were associated with potentially inappropriate testing for VDD. With the medical care of medically underserved populations being of immense importance to public health, this study provided valuable information for targeting interventions to enhance the correct allocation of vitamin D testing to women, especially those from medically underserved areas.

## Figures and Tables

**Figure 1 jcm-12-00993-f001:**
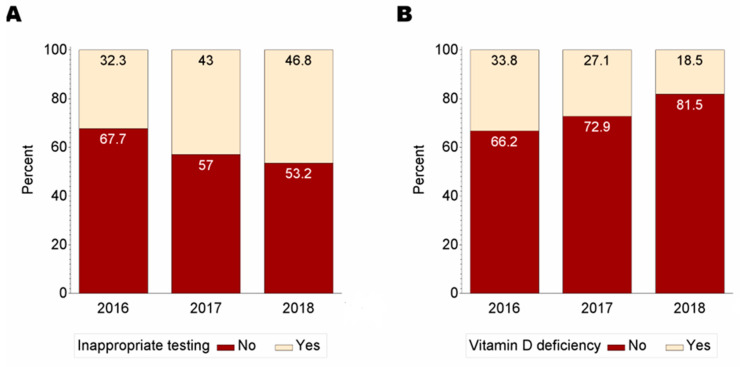
The proportion of (**A**) potentially inappropriate testing for vitamin D deficiency and (**B**) vitamin D deficiency among participants during the period of observation.

**Table 1 jcm-12-00993-t001:** Characteristics of patients according to vitamin D testing status.

	Vitamin D Test Performed		
Characteristics, %	No (n = 19,654)	Yes (n = 1753)	*p* Value
Age, years			<0.001
18–34	45.1	7.9	
35–64	32.9	41.2	
≥65	22.0	50.9	
Year of admission			0.014
2016	38.4	41.8	
2017	32.8	31.7	
2018	28.8	26.5	
Race			0.003
White	83.0	84.9	
Black	6.3	7.0	
Other	4.7	3.0	
Unknown	6.0	5.0	
Type of insurance			<0.001
Public	57.6	75.6	
Private	31.6	15.2	
Uninsured	10.8	9.2	
Marital status			<0.001
Single	38.8	27.2	
Married	40.7	38.8	
Divorced/separated/widowed	16.6	31.0	
Other/unknown	3.9	3.1	
Vitamin D supplement intake	4.5	35.7	
Prevalent medical conditions			
Cardiovascular disease	24.9	48.7	<0.001
Diabetes	6.5	13.7	<0.001
Cancer	4.0	5.8	0.001
Chronic obstructive pulmonary disease	3.2	5.2	<0.001
Clinical indications for vitamin D testing	75.7	60.5	<0.001
Lymphoma	0.2	0.5	0.138
Osteoporosis	0.1	0.3	0.006
Chronic kidney disease	1.4	16.5	<0.001
Liver disease	2.2	4.1	<0.001
Pancreatic insufficiency	0.4	0.6	0.260
Pregnancy-related hospitalization	41.4	2.2	<0.001
Obesity	52.9	45.8	<0.001
Malabsorption syndromes	0.6	0.9	0.109

**Table 2 jcm-12-00993-t002:** Characteristics of participants according to potentially inappropriate testing for vitamin D deficiency.

	Potentially Inappropriate Testing		
Characteristics, %	No (n = 1059)	Yes (n = 693)	*p* Value
Age, years			<0.001
18–34	9.5	5.5	
35–64	49.5	28.6	
≥ 65	41.0	65.9	
Year of admission			<0.001
2016	46.8	34.2	
2017	29.8	34.4	
2018	23.3	31.4	
Race			<0.001
Non-White	20.9	6.2	
White	79.1	93.8	
Type of insurance			0.002
Public	72.7	80.1	
Private	17.3	12.0	
Uninsured	10.0	7.9	
Marital status			0.007
Single	29.7	23.4	
Married	39.0	38.5	
Divorced/separated/widowed	28.7	34.3	
Other/unknown	2.6	3.8	
Prevalent medical conditions			
Cardiovascular disease	55.8	37.8	<0.001
Diabetes	16.9	8.9	<0.001
Cancer	5.9	5.5	0.753
Vitamin D supplement intake			0.022
No	66.5	61.0	
Yes	33.5	39.0	
Vitamin D deficiency			<0.001
No	66.6	81.2	
Yes	33.4	18.8	
Died during inpatient stay			<0.001
No	84.4	92.1	
Yes	15.6	7.9	

**Table 3 jcm-12-00993-t003:** Factors’ association with potentially inappropriate testing for vitamin D deficiency among participants without indications for testing.

Variables	OR (95% CI)	*p* Value
Age group		<0.001
18–34 years	1	
35–64 years	2.10 (1.45–3.06)	
≥65 years	3.07 (2.05–4.59)	
Race		<0.001
Non-White	1	
White	2.71 (1.95–3.78)	
Marital status		0.758
Single	1	
Married	0.94 (0.74–1.19)	
Divorced/separated/widowed	0.88 (0.68–1.12)	
Other/unknown	0.89 (0.56–1.44)	
Type of insurance		0.004
Private	1	
Public	1.62 (1.20–2.17)	
Uninsured	1.14 (0.79–1.65)	
Vitamin D supplement intake	7.05 (5.82–8.54)	<0.001
Cardiovascular disease	0.75 (0.63–0.90)	0.002
Diabetes	1.06 (0.78–1.43)	0.723
Cancer	0.71 (0.50–1.03)	0.068

All variables in the table were included in the multivariable model. CI: confidence interval, OR: odds ratio.

## Data Availability

The datasets generated and analyzed during the current study are not publicly available due to data protection considerations but are available from the corresponding author on reasonable request and approval by an institutional review board.

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
