# Peer review of "Factors Associated with Potentially Inappropriate Screening for Vitamin D Deficiency among Women in Medically Underserved Regions of West Texas"

_jcm, 2023, doi:10.3390/jcm12030993_

Round 1
Reviewer 1 Report
Thank you for the opportunity to review your paper. I think this is an excellent aim to address what is a pressing resource issue.
In the introduction you do not reference Avenell et al's Cochrane review which then demonstrated the research fraud of Sato which effectively removed almost all the papers showing vitamin D was of benefit outside of rickets and osteomalacia. Therefore, although you say papers recommend vitamin D testing in high risk groups this is potentially overkill too as no one has ever shown benefit really in any of these cohort (except maybe the institutionalized, very frail elderly). Also you don't mention analytical issues with measuring vitamin D e.g. immunoassay, poor recovery of D2, interference, expense, difficulty, 25D versus 1,25 D etc. I appreciate you saying that testing should be limited but one, I think at least, can still criticise those recommendations as non-evidence based and still likely encouraging over medicalisation. I know though this wasn't what you were testing, you were looking at the current guidelines now whether the current guidelines are appropriate.
In the methods, you don't mention how vitamin D was measured (and taken e.g. specimen tube etc)? We must know the method, the CV, accreditation status as per ISO standards etc. Particularly for steroid hormones which are incredibly hard to measure. I am also unclear, apologies if missed it, but when were people bled - before or during admission etc? It is just that Vit D is an acute phase protein (Waldron JL, Ashby HL, Cornes MP, Bechervaise J, Razavi C, Thomas OL, Chugh S, Deshpande S, Ford C, Gama R. Vitamin D: a negative acute phase reactant. J Clin Pathol. 2013 Jul;66(7):620-2. doi: 10.1136/jclinpath-2012-201301. Epub 2013 Mar 1. PMID: 23454726.) so you would expect it to be lower in the ones who died etc if we take death as a marker of bad acute phase etc? If these were in hospital tests you can look to see if 'appropriate' but you will need to be careful with interpreting 'low' concentrations. If taken in hospital, and given it is an acute phase protein, is it ever appropriate to measure in the acutely ill?
I like the fact you include being on a supplement as a reason for inappropriate testing.
In the discussion you talk about the recognized benefits of vitamin D- I would check your data but I believe that outside of VDD there are no benefits. The prevalence of VDD is going up but not rickets or osteomalacia therefore we are creating an epidemic based on a test, not patient symptoms/consequences that can be altered. However the rest of the discussion is really good and I enjoyed your thoughts about the very real impact this is having on health care and the patient.
Author Response
Reviewer 1
- In the introduction you do not reference Avenell et al's Cochrane review which then demonstrated the research fraud of Sato which effectively removed almost all the papers showing vitamin D was of benefit outside of rickets and osteomalacia. Therefore, although you say papers recommend vitamin D testing in high risk groups this is potentially overkill too as no one has ever shown benefit really in any of these cohort (except maybe the institutionalized, very frail elderly).
Response
We thank the reviewer for the kind words and the thorough review of our manuscript. As requested, we now include information in the introduction, including the Cochrane review article, that indicates that there is little to no association between vitamin D supplementation and health outcomes (lines 33 to 35)
- Also you don't mention analytical issues with measuring vitamin D e.g. immunoassay, poor recovery of D2, interference, expense, difficulty, 25D versus 1,25 D etc. I appreciate you saying that testing should be limited but one, I think at least, can still criticise those recommendations as non-evidence based and still likely encouraging over medicalisation. I know though this wasn't what you were testing, you were looking at the current guidelines now whether the current guidelines are appropriate.
Response
We thank the reviewer for bring up this important information. As the reviewer duly acknowledged, this was not part of the aim of our study so we decided not to discuss such information.
- In the methods, you don't mention how vitamin D was measured (and taken e.g. specimen tube etc)? We must know the method, the CV, accreditation status as per ISO standards etc. Particularly for steroid hormones which are incredibly hard to measure. I am also unclear, apologies if missed it, but when were people bled - before or during admission etc? It is just that Vit D is an acute phase protein (Waldron JL, Ashby HL, Cornes MP, Bechervaise J, Razavi C, Thomas OL, Chugh S, Deshpande S, Ford C, Gama R. Vitamin D: a negative acute phase reactant. J Clin Pathol. 2013 Jul;66(7):620-2. doi: 10.1136/jclinpath-2012-201301. Epub 2013 Mar 1. PMID: 23454726.) so you would expect it to be lower in the ones who died etc if we take death as a marker of bad acute phase etc? If these were in hospital tests you can look to see if 'appropriate' but you will need to be careful with interpreting 'low' concentrations. If taken in hospital, and given it is an acute phase protein, is it ever appropriate to measure in the acutely ill?
Response
We mentioned in the methods under definition of variables that vitamin D status measured at the first time of admission was used in this study. The question of the appropriateness of measuring vitamin D in acutely ill patients is indeed salient and another reason why our study of inappropriate testing of vitamin D deficiency is of importance as it shows patterns of testing in such settings. Unfortunately, the hospital did not provide methods by which vitamin D were analyzed as these are usually not available in medical records. We agree that that such information is important so we have acknowledged it as a limitation in the discussion session of the manuscript.
- I like the fact you include being on a supplement as a reason for inappropriate testing.
Response
We thank the reviewer for the very kind words.
- In the discussion you talk about the recognized benefits of vitamin D- I would check your data but I believe that outside of VDD there are no benefits. The prevalence of VDD is going up but not rickets or osteomalacia therefore we are creating an epidemic based on a test, not patient symptoms/consequences that can be altered. However the rest of the discussion is really good and I enjoyed your thoughts about the very real impact this is having on health care and the patient.
Response
Again, we thank the reviewer for the very kind words. We have revised some of our sentences about the benefit of vitamin D levels in the etiology of health outcomes by stressing that the evidence of benefits is from observational studies, with experimental studies showing little or no benefits of vitamin D supplementation on health outcomes. (Lines 33 to 35; 186 to 190)
Reviewer 2 Report
Well written
1.Kindly mention pointwise-- the which Patients were considered to be at high risk for VDD
and except those are the target group
2. Results tables are there need graphical representation to understand or notice the gap clearly
3. How medically underserved regions not related with ---- "Black race or Hispanic ethnicity, were obese or reported any of the following conditions: pregnant and/or lactating (Z33, 92 Z34, O00-O9A); rickets (E55); osteomalacia (M83); osteoporosis (M81); adults aged > 50 93 years with history of falls or nontraumatic fractures (W18.30XA, Z91.81); chronic kidney disease (N18, N19); liver disease (K70-K77); pancreatic Insufficiency (K86); malabsorptive syndromes (V44, K90) such as cystic fibrosis (E84); inflammatory bowel disease (K50, 96 K51); Crohn's disease (K50), celiac disease (K90.0), gastric bypass surgery (Z98.84) and radiation enteritis (K52.0, K62.7); hyperparathyroidism (E21.2); lymphoma (C81-C88); and granuloma-forming disorders namely sarcoidosis (D86), tuberculosis (A15-A19), histoplasmosis (B39), coccidiomycosis (B38), and berylliosis (J63.2)" and considered as potentially inappropriate screening for VDD-- Give explanation
Author Response
Reviewer 2
- Kindly mention pointwise-- the which Patients were considered to be at high risk for VDD and except those are the target group
Response
We thank the reviewer for the kind words and the thorough review of our manuscript. As requested, we mention in the results section of the manuscript that “Almost 74% of women admitted had a medical condition or race and ethnicity that was considered to be an indication for a high risk for VDD.” (Lines 129 to 132)
- Results tables are there need graphical representation to understand or notice the gap clearly.
Response
We thank the reviewer for this comment. We have provided graphical representation for the proportion of inappropriate testing over time as well as the proportion of vitamin D deficiency over time in Figure 1.
- How medically underserved regions not related with ---- "Black race or Hispanic ethnicity, were obese or reported any of the following conditions: pregnant and/or lactating (Z33, 92 Z34, O00-O9A); rickets (E55); osteomalacia (M83); osteoporosis (M81); adults aged > 50 93 years with history of falls or nontraumatic fractures (W18.30XA, Z91.81); chronic kidney disease (N18, N19); liver disease (K70-K77); pancreatic Insufficiency (K86); malabsorptive syndromes (V44, K90) such as cystic fibrosis (E84); inflammatory bowel disease (K50, 96 K51); Crohn's disease (K50), celiac disease (K90.0), gastric bypass surgery (Z98.84) and radiation enteritis (K52.0, K62.7); hyperparathyroidism (E21.2); lymphoma (C81-C88); and granuloma-forming disorders namely sarcoidosis (D86), tuberculosis (A15-A19), histoplasmosis (B39), coccidiomycosis (B38), and berylliosis (J63.2)" and considered as potentially inappropriate screening for VDD-- Give explanation.
Response
We apologize that we do not understand this request of the reviewer. We did not study the relation of these conditions that are indications for vitamin D deficiency with medically underserved population so it is hard for us to provide an explanation for why they are not related